
# Measurement report: Atmospheric fluorescent bioaerosol concentrations measured during 18 months in a coniferous forest in the south of Sweden

Madeleine Petersson Sjögren[1], Malin Alsved[1], Tina Šantl-Temkiv[2], Thomas Bjerring Kristensen[3,4], Jakob Löndahl[1]

[1]Department of Design Sciences, Lund University, Lund, Sweden
[2]Department of Biology, Microbiology Section and iCLIMATE Aarhus University Interdisciplinary Centre for Climate Change, Aarhus University, Aarhus, Denmark
[3]Department of Physics, Lund University, Lund, Sweden
[4]Force Technology, 2605 Brøndby, Denmark

*Correspondence to*: Jakob Löndahl (Jakob.Londahl@design.lth.se)

**Abstract.** Biological aerosol particles affect human health, are essential for microbial- and gene dispersal, and have been proposed as important agents for atmospheric processes. However, the abundance and size distributions of atmospheric biological particles are largely unknown. In this study we used a laser-induced fluorescence instrument to measure fluorescent biological aerosol particle (FBAP) concentrations for 18 months (October 2020 – April 2022) at a rural, forested site in Sweden. The aim of this study was to investigate FBAP number concentrations ($N_{FBAP}$) over time and analyze their relationship to meteorological parameters.

The $N_{FBAP}$ was highest in the summer and lowest in winter, exhibiting a ~3-fold difference between these seasons. The median $N_{FBAP}$ was 0.0050, 0.0025, 0.0027 and 0.0126 cm$^{-3}$ in fall, winter, spring, and summer, respectively, and constituted ~0.1-0.5 % of the total supermicron particle number. The $N_{FBAP}$ were dominated by the smallest measured size fraction (1-3 µm), suggesting that the main portion of the biological particles measured were due to single bacterial cells, fungal spores, and bacterial agglomerates. The $N_{FBAP}$ were significantly correlated with increasing air temperature (P<0.01) in all seasons. For most of the campaign $N_{FBAP}$ was seen to increase with wind speed (P<0.01), while the relationship with relative humidity was for most part of the campaign nonsignificant (46 %) but to a large part (30 %) negative (P<0.05). Our results indicate that $N_{FBAP}$ were highest during warm and dry conditions when wind speeds were high, suggesting that a major part of the FBAP in the spring and summer were due to mechanical aerosol generation and release mechanisms. In the fall, relative humidity may have been a more important factor for bioaerosol release. This is one of the longest time series of atmospheric FBAP, which are highly needed for estimates of bioaerosol background concentrations in comparable regions.



## 1 Introduction


Primary biological aerosol particles (PBAPs), also called bioaerosols, make up a diverse set of particles. They constitute airborne fungal cells and spores, bacteria, pollen, plant, and animal debris as well as biomolecules, present on their own or attached to other particles. Bioaerosols are emitted into the atmosphere, from every region and ecosystem of the planet, and range in size from a few nm to 100 μm (Despres et al., 2012; Womack et al., 2015)


They can influence climate and potentially the hydrological cycle by acting as ice nucleating particles (INPs) (Pöschl et al., 2010; Prenni et al., 2009b; Hill et al., 2016). Diverse microorganisms, including bacteria, fungi, microalgae, pollen, and lichens are known to produce high-temperature ice nucleating compounds and have therefore been proposed to influence cloud formation (Diehl et al., 2001; Lohmann and Feichter, 2005; Bowers et al., 2009; Pratt et al., 2009; Prenni et al., 2009a).

Bioaerosols transmit pathogens, spread diseases, and can have toxic, infectious and allergic effects in humans, animals, crops, and ecosystems. They can cause human respiratory diseases when inhaled or deposited in mucus in the eyes (Franze et al., 2005; Lacey and Dutkiewicz, 1994; Kim et al., 2018; Brown and Hovmøller, 2002). Despite their large size, bioaerosols are aerodynamically buoyant and can spread thousands of kilometers across land and oceans (Griffin et al., 2007; Burrows et al., 2009a). Atmospheric dispersal of bioaerosols depends on particle size and meteorological parameters, including wind,

humidity, temperature, convection, and turbulence (Norros et al., 2014; Madelin, 1994; Jones and Harrison, 2004).

Bioaerosols have traditionally been collected and measured with offline techniques, but continuous online measurements are needed to increase time resolution and improve understanding on variability driven by meteorological factors as well as diurnal, or seasonal cycles (Huffman et al., 2019; Šantl-Temkiv et al., 2020). The offline methods incorporate filter collection,

impactors, impingers and electrostatic precipitators. While these measurements promote specificity by determining the identity and properties of bioaerosols, they lack sensitivity with respect to time and size resolution. Progress in the detection of bioaerosols with higher time resolution has been made by utilizing laser-induced fluorescence (LIF) (Hill et al., 1995).

LIF is based on the intrinsic fluorescence light emission of organic molecules that contain fluorophores such as amino acids,

coenzymes, vitamins, biopolymers and cell wall compounds (Li et al., 2019). Instruments with LIF provide continuous and quantitative measurements of fluorescent biological aerosol particles (FBAPs). While this method cannot offer specificity or imaging capabilities comparable to downstream laboratory analysis, it offers real-time detection and discrimination of bioaerosols with high time and size resolution. Such real-time measurements promote understanding of the temporal variations in abundance, sources, and emission factors of bioaerosols. Model studies emphasize the need for more and continuous data



to constrain models on emission and transport of atmospheric bioaerosols (Burrows et al., 2009b; Burrows et al., 2009a; Heald and Spracklen, 2009).

Despite the wide-ranging influence of bioaerosols on climate, agriculture and public health, long-term real-time data on bioaerosols are limited. Table 1 summarizes the 8 continuous real-time FBAP number concentration ($N_{FBAP}$) measurements

longer than 4 weeks that have been made. These studies were conducted in a wide range of environments (remote, rural, urban, tropical, and high-altitude) in Asia, North America, South America, and Europe, with measurement periods ranging from a few weeks to a maximum of 20 months (Huffman et al., 2010; Huffman et al., 2012; Huffman et al., 2013; Schumacher et al., 2013; Toprak and Schnaiter, 2013; Saari et al., 2015; Gosselin et al., 2016; Valsan et al., 2016). The most common commercially available instruments used for LIF measurement of $N_{FBAP}$ are the ultraviolet aerodynamic particle sizer (UV-

APS; TSI Inc. St. Paul, Minnesota, USA) and the wideband integrated bioaerosol sensor (WIBS; DMT, Longmont, Colorado, USA). The UV-APS was discontinued in 2014. Other LIF-instruments include the BioScout (Environics Oy, Finland) and the BioTrak real-time viable particle counter (TSI Inc. St. Paul, Minnesota, USA). In the studies here listed $N_{FBAP}$ ranged between 0.015 cm$^{-3}$ measured in the winter in Finland and Colorado (Schumacher et al., 2013), to 0.073 cm$^{-3}$ measured in the Amazonian tropical rainforest (Huffman et al., 2012). When seasonal variations were addressed, $N_{FBAP}$ were clearly highest in

the summer and lowest in the winter (Schumacher et al., 2013; Toprak and Schnaiter, 2013). The $N_{FBAP}$ made up between 2 and 24 % of the total supermicron (particle diameters=1-10μm) concentration (Valsan et al., 2016; Huffman et al., 2012). Several studies showed that increased relative humidity (RH) was positively correlated with FBAP concentrations, while air temperature (AT) was negatively correlated (Huffman et al., 2010; Huffman et al., 2012; Schumacher et al., 2013; Toprak and Schnaiter, 2013; Saari et al., 2015; Valsan et al., 2016). In some cases, rain events were associated with significant increases

in $N_{FBAP}$ (Huffman et al., 2012; Schumacher et al., 2013), but the pattern is inconsistent (Toprak and Schnaiter, 2013). No clear patterns have been observed for the effect of wind speed and wind direction on $N_{FBAP}$, although several studies have addressed it (Huffman et al., 2010; Schumacher et al., 2013; Valsan et al., 2016). Clear 24-h cycles (also referred to as diurnal cycles) have been identified (Huffman et al., 2010; Huffman et al., 2012; Schumacher et al., 2013; Toprak and Schnaiter, 2013; Saari et al., 2015; Valsan et al., 2016). An extended version of Table 1, also including shorter LIF-FBAP measurements, is included

in the supplementary material.

The aim of this study was to investigate short- and long-term drivers behind FBAP, as a proxy for PBAP, in rural boundary layer air for a period covering all seasons in the Southern Sweden. We used a LIF instrument (BioTrak, TSI Inc. St. Paul, Minnesota, USA) for counting and sizing of FBAP during 18 months between October 2020 and April 2022 at the Hyltemossa

Aerosols, Clouds, and Trace gases Research InfraStructure (ACTRIS) and integrated carbon observation system (ICOS) station. Thus, this is one of the longest multi-month ambient measurement studies for real-time bioaerosol detection using a LIF-instrument.



**Table 1: Summary of previous long-term (>4 weeks) continuous measurements of $N_{FBAP}$ with real-time detection, with identified associations and correlations with meteorological parameters and cycles in $N_{FBAP}$. When no average $N_{FBAP}$ or $N_{FBAP}/N_{TAP}$ was reported, this is indicated by a hyphen. When both mean and median values were reported, they are listed as mean/median.**

| Location | Land use | Instrument | Measurement period | Season(s) | Average $N_{FBAP}$ (cm⁻³) Mean/ Median | $N_{FBAP}$ % of super micron parti cles | Associations between FBAP and meteorology observed | $N_{FBAP}$-cycles |
|---|---|---|---|---|---|---|---|---|
| Mainz, Germany[1] | Semi-urban | UV-APS | 4 months: Aug-Dec 2006 | Fall Winter | 0.03 | 4 | FBAP increased with RH<br><br>No correlation with WD. | 24-h cycle with max early/mid-morning |
| Amazon, Brasil[2] | Tropical rainforest | UV-APS | 5 weeks: Feb-Mar 2008 | Rain season | 0.073 | 24 | FBAP increased with RH, FBAP decreased with AT and heavy rain was associated with FBAP increases. | 24-h cycle with max in the night |
| Colorado, USA[3] | Semi-arid, rural forest | UV-APS WIBS-4 | 5 weeks: Jul-Aug 2011 | Summer | - | - | FBAP increased during rain | - |
| Hyytiälä, Finland[4] | Rural forest | UV-APS | 20 months: Aug 2009-April 2011 | Spring Summer Fall Winter | 0.015 0.046 0.027 0.004 | 4.4 13 9.8 1.1 | FBAP scaled with RH in summer in both locations. In Finland, at RH>82 % FBAP decreased. | 24-h cycle with max evening/ night for all seasons. |
| Colorado, USA[4] | Semi-arid, rural forest. | | 11 months: Jul 2011- May 2012 | Spring Summer Fall Winter | 0.015 0.030 0.017 0.0053 | 2.5 8.8 5.7 3.0 | FBAP increased upon rain events, FBAP increased with AT over seasons. No pattern observed for wind speed or wind direction. | |
| South-Western Germany[5] | Semi-rural | WIBS-4 | 1 year: April 2010-April 2011 | Spring<br>Summer<br>Fall<br>Winter<br>Full year: | 0.029/0.024<br>0.046/0.040<br>0.029/0.023<br>0.019/0.017<br>0.031/0.025 | 7/5<br>10/9<br>7/6<br>3/4<br>7/5 | FBAP positively correlated with RH. No other correlations found between FBAP and meteorology. | 24-h cycle with max late evening/ early morning. |
| Helsinki, Finland[6] | Suburban and urban | BioScout UV-APS | 3 weeks: Feb 2012<br><br>9 weeks: Jun-Aug 2012 | Winter<br><br>Summer | 0.010<br><br>0.028 | 5<br><br>23 | | 24-h cycle with max in the night during summer. |
| Colorado, USA[7] | Semi-arid rural forest. | UV-APS WIBS-3 | 5 weeks: Jul-Aug, 2014 | Summer | - | - | - | - |



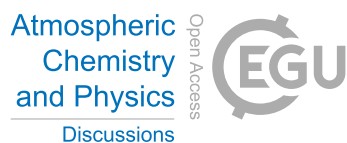

| Munnar, India[8] | Tropical, high altitude. | UV-APS | 11 weeks: June-Aug, 2014 | Monsoon and winter* | 0.2 | 2 | FBAP strongly dependent on WD, increased with RH, decreased with AT and decreased with WS. | 24-h cycle with max at high RH and low AT. |
|---|---|---|---|---|---|---|---|---|

[1,2,3] Huffman et al. (2010, 2012, 2013); [4] Schumacher et al (2013); [5] Toprak and Schnaiter (2013); [6]Saari et al. (2015); [7]Gosselin et al. (2016); [8]Valsan et al. (2016).
*Results were only reported for the Monsoon period and not for the winter
Abbreviations: relative humidity RH; air temperature AT; wind direction WD; wind speed WS; total aerosol particles TAP; fluorescent biological aerosol particles FBAP.

## 2 Methods

### 2.1 Hyltemossa research site

The measurements were performed at the Hyltemossa research station from 6 October 2020 until 1 April 2022. The site is a combined aerosols, clouds, and trace gases research infrastructure (ACTRIS) and integrated carbon observation system (ICOS) station, situated at lat: 56° 5' 52" N and long: 13°25' 8" E. The sampling site, established in 2014, is located in a managed coniferous forest. Sections of the forest are clear-cut every 50 years and the trees grow about 35 meters in 100 years. The vegetation is dominated by Norway spruce (*Picea abies*) with a low fraction of birch trees (*Betula sp.*) and some Scots pine (*Pinus sylvestris*). The average tree canopy height at the sampling site is ~20 m. The forest floor is covered by a thick moss layer and very little shrub life grows under the trees.

### 2.2 Fluorescent biological aerosol measurement

We used a BioTrak® real-time viable particle counter (TSI Inc., US) for continuous measurement of atmospheric $N_{FBAP}$ and total supermicron (1-12 µm) aerosol particle number concentration ($N_{TAP}$). The instrument combines an optical particle counter (OPC) system with a fluorescence detector to determine if a particle is biological based on a built-in discrimination algorithm. The BioTrak® particle counter has a sample flow of 28.3 Lmin$^{-1}$. The sample airflow first passes the OPC where size and total particle number concentration are determined using single particle light scattering from a 660 nm laser. The BioTrak OPC has 6 size channels with lower cut-off diameters 0.5, 0.7, 1, 3, 5 and 10 µm. Particles are concentrated and the airflow is lowered down to 1 Lmin$^{-1}$ with a concentrator to enable measurement of low intensity fluorescence with the LIF detector. Most of the smallest particles (1< µm) will follow the exhaust flow in the virtual impactor, hence, primarily particles >1 µm reach the LIF detector. In the LIF detector, single particles are illuminated by a 405 nm collimated laser and three independent optical signals are collected. The first signal is scattered light detected with an avalanche photo detector (APD) that gives the particle size. The second and third signals are the emitted fluorescence light that is collimated and separated into two wavelength bands, at 405-500 nm and 500-600 nm, respectively, collected with two separate photomultiplier tubes. The three independent signals are converted to electrical signals, digitized, and fed to a detection algorithm that classifies the particle as biological (fluorescent) or not. This is the first report on use of the BioTrak for measurement of bioaerosols.



Ambient air was drawn through a $PM_{10}$ inlet designed for an airflow of 38.3 Lmin$^{-1}$, (compare with airflow of 28 Lmin$^{-1}$ in BioTrak), which made the cutoff diameter increase from 10 μm to ~12 μm. The BioTrak® instrument was placed inside the Hyltemossa research station with a vertical ~4 m long sampling tube to the inlet located about 5 m above ground level. The $N_{FBAP}$ and $N_{TAP}$ were sampled with a frequency of 5 min. Note that in the following, we use supermicron to refer to particle sizes 1-12 μm.


The BioTrak flowrate was regularly checked with an external flow meter (TSI 4000 Series thermal mass flow meter) and was within 3 % of the given flowrate. Sampling was interrupted 1-2 hours each month when data was downloaded, and the instrument was checked and cleaned. A total of 7 days of data are missing in December 2021 (1 Dec until 8 of December) due to overloaded memory of the instrument.

**2.3 Meteorological data**

Instruments measuring air temperature, relative humidity, wind speed and wind direction are operated continuously by ICOS Sweden at the Hyltemossa research station at 30 m, 70 m, and 150 m above ground. For the comparisons with the data on biological aerosol concentrations, we used the measurements at both 30 and 70 m, which gave very similar results. In the following, we used the 70 m meteorological data for comparisons with particle data. Meteorological measurements were hourly

mean values. Precipitation was measured cumulatively every 30 minutes. Reported accuracy for the meteorological measurements made at the site were as follows: air temperature $\pm$ 0.1°C; RH $\pm$ 0.8 %; wind speed $\pm$ 2 %; wind direction $\pm$ 5° and precipitation $\pm$ 1 %.

**2.4 Data processing and statistical analysis**

In the analysis, the particle number concentrations, $N_{FBAP}$ and $N_{TAP}$, and their ratio, were averaged over days, months, and

seasons. Measurement periods were averaged into seasons as defined meteorologically by the Swedish meteorological and hydrological institute (SMHI): fall begins after 5 consecutive days with daily mean temperature below 10.0°C and 1 August is the earliest allowed date for fall to start; winter begins after 5 consecutive days with daily mean temperatures at 0.0°C or below; spring begins after 7 consecutive days with daily mean temperatures above 0.0°C with the earliest allowed date at 15$^{th}$ of February; summer begins after 5 consecutive days with daily mean temperatures above 10.0°C. In 2020, 2021 and 2022 the

seasons were as follows: Fall (October 8, 2020 - January 12, 2021), winter (January 13 – February 16, 2021), spring ( February 17 – May 7, 2021), summer (May 8 – October 5, 2021), fall (October 6 - November 25, 2021), winter (November 26, 2021 – February 15, 2022), spring (started February 15, 2022) (www.smhi.se). It should be noted that there are other ways to define the seasons. It can be argued that sunlight and/or day length would be more appropriate for studying bioaerosol drivers, since sunlight is very important for vegetation phenology that could be a main driver of bioaerosol emissions. Differences between

monthly and seasonal concentrations were assessed with Kruskal Wallis tests and post-hoc Mann-Whitney tests with

Bonferroni corrections since $N_{FBAP}$ distributions were skewed. In the following, medians were used to represent data if not otherwise stated.

To study the relationship between $N_{FBAP}$ and meteorological parameters, $N_{FBAP}$ data were binned based on meteorological parameters. For air temperature, 21 bins were constructed between -10°C and 31 °C. For relative humidity, 10 bins were constructed between 0 and 100 %. For wind speed 13 bins were constructed between 0 and 13 ms$^{-1}$. Bins that contained less than 0.1 % of the total particle counts within the given season were removed to ensure that only statistically significant observations were included in the detailed analysis. Spearman's rank coefficient ($\rho$) and Pearson's linear regression coefficient (r), with associated P-values, were used to assess the degree of association and linearity between $N_{FBAP}$ binned data and meteorological parameters. Rolling Pearson's correlation coefficients with associated P-values were calculated to assess the linear relationship between $N_{FBAP}$ throughout the full campaign period. Weekly rolling correlations were calculated for hourly mean values of meteorological parameters and $N_{FBAP}$.

Statistical distributions of $N_{FBAP}$, $N_{TAP}$ and $N_{FBAP}/N_{TAP}$ were presented as box-whisker plots, showing the arithmetic mean, the median, 25$^{th}$ and 75$^{th}$ percentiles, and 5$^{th}$ and 95$^{th}$ percentiles. Daily cycles in $N_{FBAP}$, $N_{FBAP}/N_{TAP}$ and meteorological parameters were explored by averaging the data for each hour of the day. $N_{FBAP}$ abundance differences between days and nights were assessed by distinguishing $N_{FBAP}$ based on local sunrise and sunset times. Particle mass concentrations were calculated for each size channel by multiplication with the aerodynamically equivalent sphere with the geometric midpoint diameter assuming a density of 1 gcm$^{-3}$ and a shape factor of 1, for $N_{FBAP}$, $N_{TAP}$ and $N_{FBAP}/N_{TAP}$.

Rain events were identified on an hourly basis as all consecutive hours with precipitation, equal to or higher than 0.5 mmh$^{-1}$. For each rain event, $N_{FBAP}$ was assessed right before the rain event, during, and right after. The lengths of the period before and after rainfall were defined to be the same length as the rain duration. The longest rain event recorded lasted 21 hours.

## 3 Results and discussion

### 3.1 General trends

### 3.1.1 Monthly and seasonal trends in fluorescent biological aerosol particles

The $N_{FBAP}$ exhibited clear seasonal patterns, with overall highest $N_{FBAP}$ concentrations in the summer and lowest in the winter, respectively (P<0.0001). The $N_{FBAP}$ data covered approximately two full fall seasons, two winter seasons, approximately one and a half spring and one full summer season (see Table 2). Figure 1 shows $N_{FBAP}$ at 5 min sampling resolution (green) and 7-195 day median (magenta). Seasons according to the SMHI are delimitated by vertical lines.

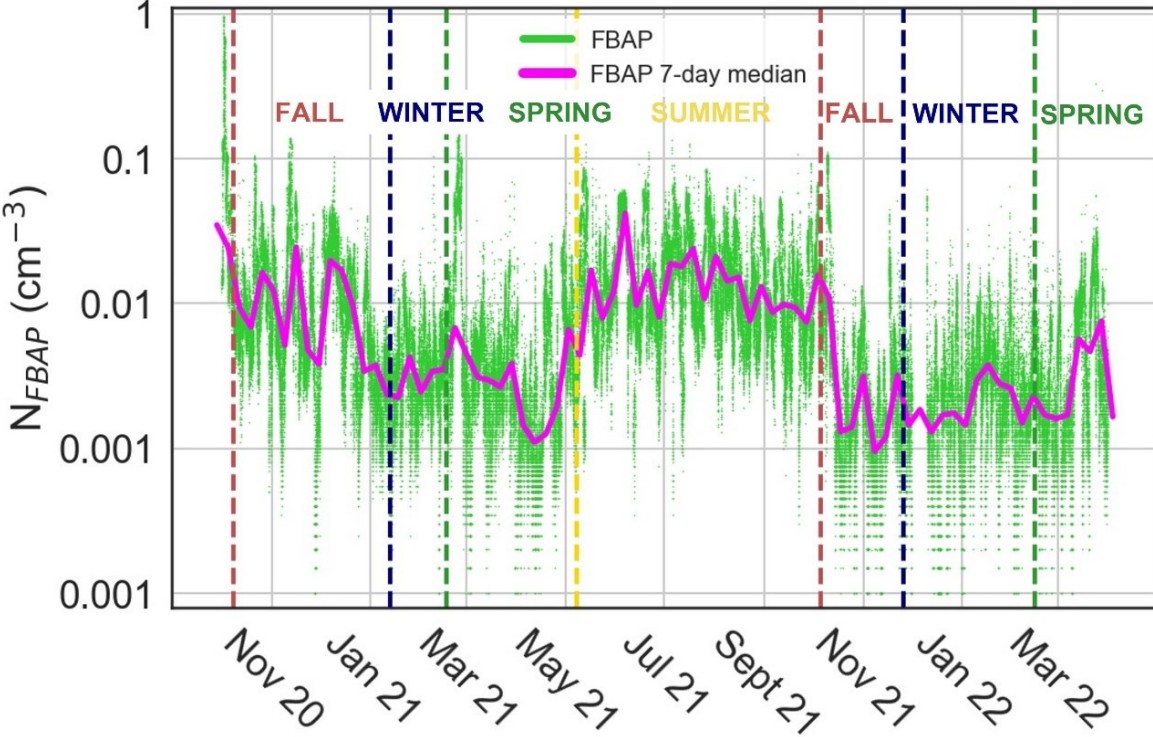

**Figure 1: Overview of fluorescent biological aerosol particle number concentration ($N_{FBAP}$) between 6 October 2020 and 1 April 2022. Green dots represent individual $N_{FBAP}$ 5 min data averages and magenta curve shows running 7-day median values. Vertical dashed lines indicate the first day of each season as identified by the Swedish Meteorological and Hydrological Institute. $N_{FBAP}$ abundance is seen to increase steeply at the intersection of spring and summer (~ 8 May 2021) and decrease steeply in the beginning of fall (~6 October 2021). The highest $N_{FBAP}$ concentrations were measured in the summer and the lowest concentrations were measured in the winter periods.**






**Table 2: Seasonal and full campaign comparisons between fluorescent biological aerosol particle number concentration (N_FBAP), total supermicron aerosol particle number concentration (N_TAP) and biological aerosol particle ratio N_FBAP/TAP. Particle concentrations are shown as arithmetic means and medians with associated standard deviations ± SD and inter-quartile range (IQR) in parenthesis for median values, over entire seasons and for the full campaign. Average temperatures, relative humidity, wind speeds and precipitation are also listed and the diel N_FBAP peak hour of the day per season.**

| | Fall | Winter | Spring | Summer | Full campaign |
|---|---|---|---|---|---|
| Duration | 08 Oct 2020 – 12 Jan 2021 and 06 Oct 2021 – 25 Nov 2021 | 13 Jan 2021 – 16 Feb 2021 and 26 Nov 2021 – 14 Feb 2022 | 17 Feb 2021 – 07 May 2021 and 15 Feb 2022 – 1 Apr 2022 | 08 May 2021– 05 Oct 2021 | 1 Oct 2020 – 1 Apr 2022 |
| No. of days | 148 | 153 | 80 | 156 | 537 |
| Mean $N_{FBAP}$ (cm$^{-3}$) | 0.0093±0.0128 | 0.0038±0.0049 | 0.0074±0.0176 | 0.0211±0.0429 | 0.0109±0.0261 |
| Median $N_{FBAP}$ (cm$^{-3}$) | 0.0050 (0.0094) | 0.0025 (0.0033) | 0.0027 (0.0047) | 0.0126 (0.0154) | 0.0051 (0.0097) |
| Mean $N_{TAP}$ (cm$^{-3}$) | 4.96 ±4.79 | 3.42 ±3.64 | 5.37 ±6.23 | 4.15 ±5.53 | 4.35 ±5.03 |
| Median $N_{TAP}$ (cm$^{-3}$) | 3.62 (5.35) | 2.07 (3.65) | 3.02 (5.87) | 2.96 (2.68) | 2.85 (4.03) |
| Mean $N_{FBAP}/N_{TAP}$ | 0.0030 ±0.0034 | 0.0020 ±0.0023 | 0.0017 ±0.0024 | 0.0062 ±0.0050 | 0.0035 ±0.0040 |
| Median $N_{FBAP}/N_{TAP}$ | 0.0019 (0.0036) | 0.0014 (0.0021) | 0.0011 (0.0020) | 0.0047 (0.0065) | 0.0021 (0.0037) |
| Diel $N_{FBAP}$ peak (hour of the day) | 21:00 | 15:00 | 18:00 | 16:00 | 18:00 |
| Mean Temperature (°C) | 6.1 ±3.8 | -0.1 ±4.3 | 4.3 ± 3.4 | 15.2 ± 4.1 | 7.5 ± 7.0 |
| Median Temperature (°C) | 6.6 (5.3) | -0.1 (5.6) | 3.9 (4.3) | 15.0 (5.1) | 7.0 (10.5) |
| Mean RH (%) | 93 ± 11 | 91.2 ± 14.9 | 72 ±26 | 76 ± 20 | 84 ±20 |
| Median RH (%) | 100 (12) | 99.9 (13.6) | 79 (51) | 78 (33) | 92 (27) |
| Mean Wind speed (ms$^{-1}$) | 5.2 ±2.0 | 5.5 ±2.3 | 5.3 ±1.9 | 4.8 ±1.8 | 5.2 ±2.0 |
| Median Wind speed (ms$^{-1}$) | 5.0 (2.9) | 5.3 (3) | 5.2 (2.2) | 4.5 (1.9) | 5.0 (2.1) |
| Mean Wind direction (°) | South | South-Southwest | Southwest | South | South-Southwest |
| Median Wind direction (°) | South | South-Southwest | West-Southwest | South-Southwest | South-Southwest |
| Number of rain events | 90 | 44 | 35 | 64 | 233 |
| Mean cumulative precipitation per rain event (mm) | 3.87 ±11.14 | 3.43 ± 3.84 | 3.91 ± 5.86 | 5.60 ±7.24 | 4.98 ±17.44 |
| Median cumulative precipitation per rain event (mm) | 1.35 (2.16) | 1.77 (3.28) | 2.25 (3.28) | 3.07 (5.65) | 1.80 (3.72) |






The median monthly $N_{FBAP}$ concentration over the full campaign was 0.005 cm$^{-3}$ but varied significantly (P<0.0001) over the full campaign. Monthly and seasonal distributions of $N_{FBAP}$, $N_{TAP}$ and $N_{FBAP}/N_{TAP}$ are shown in Fig. 2 and Fig. 3. The monthly

median varied by a factor of ~13, from a minimum of 0.0015 cm$^{-3}$ in December 2021 to a maximum of 0.019 cm$^{-3}$ in July 2021. The period with highest $N_{FBAP}$ abundance was initiated by a steep increase in May, where $N_{FBAP}$ increased by a factor of ~8 over a few weeks. The period with high $N_{FBAP}$ ended with a steep decrease over a few weeks between November and December in 2021, when the concentration decreased by a factor of ~8. A similar decrease of a factor of ~4 was observed in the winter 2020. $N_{FBAP}$ varied a lot over the summer months but was more stable during fall and winter when the concentrations

were lower. The overall $N_{FBAP}$ agreed with other long-term measurements conducted in Finland, USA, Germany, France, India, China (Huffman et al., 2010; Schumacher et al., 2013; Toprak and Schnaiter, 2013; Valsan et al., 2016; Yu et al., 2016). The $N_{FBAP}$ concentrations in winters were as expected lower compared to other seasons. Cold temperatures, lowered biological activity, low absolute humidity and snow coverages are suggested to cause a lower generated amount of bioaerosols in winter, but also reduce bioaerosols' abilities to be lofted into the air (Schumacher et al., 2013; Toprak and Schnaiter, 2013; Saari et

al., 2015; Huffman et al., 2010). At the beginning of fall, spores from fungi are dispersed in the air, which increases the fall $N_{FBAP}$ although temperatures decreased (Schumacher et al., 2013; Toprak and Schnaiter, 2013).

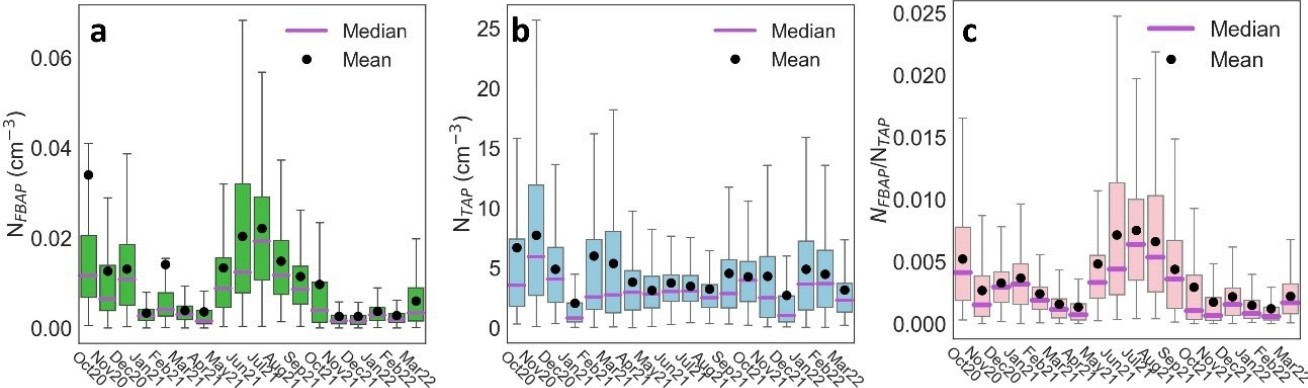

**Figure 2. Statistical distribution of monthly fluorescent biological aerosol particle number concentration (left), total supermicron**
**aerosol particle number concentration (middle) and their ratio (right) as box-whisker plots. Lower and upper limits of each box represent 25$^{th}$ and 75$^{th}$ percentiles, respectively. Vertical bars at the end of lower and upper vertical bars represent 5$^{th}$ and 95$^{th}$ percentiles, respectively. Outliers were removed from the plots to make them easier to read.**





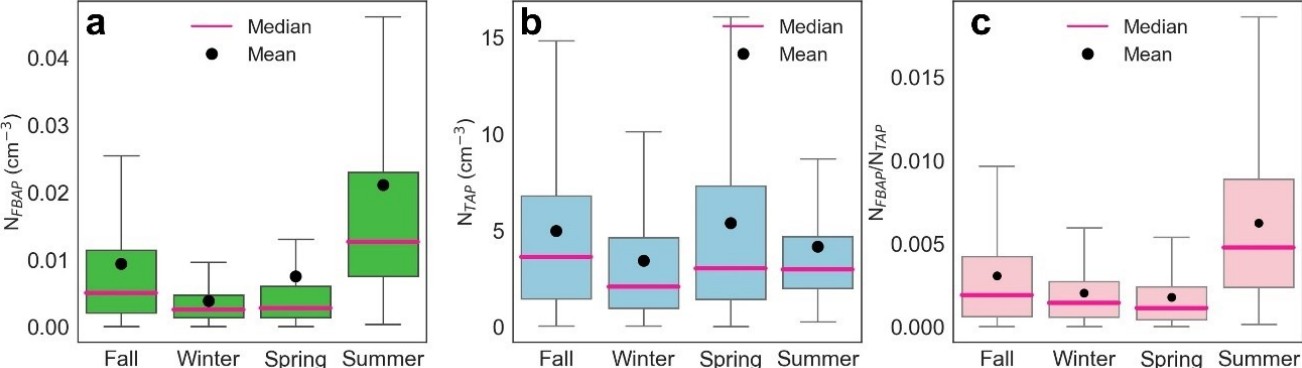

**Figure 3. Seasonal statistical distribution of fluorescent biological aerosol particle number concentration (left), total supermicron aerosol particle number concentration (middle) and their ratio (right) as box-whisker plots. Lower and upper limits of box represent 25th and 75th percentiles, respectively. Vertical bars at the end of lower and upper vertical bars represent 5th and 95th percentiles, respectively. Outliers were removed from the plots to make them easier to read.**

Not all biological material will have a sufficiently strong fluorescence signal to be detected as FBAP. This is due to that the fluorescence signal is a function of the concentration of fluorophores in PBAP, the ability to be excited by the laser of the instrument and the presence of opaque or absorbing material that may fluoresce only very weakly. Moreover, the detection threshold of ~1 μm biases the reported FBAP low. Therefore, fluorescence measurements, such as the ones made here, have the risk of underestimating the total biological material present, and the $N_{FBAP}$ measurements should be viewed as lower limits of PBAP. This was also suggested by Huffman et al. when using a UV-APS for FBAP measurements (Huffman et al., 2012; Huffman et al., 2010). Also, further investigations are needed to better understand the response of the BioTrak to different types of biogenic aerosol particles and to quantity potential interferences with non-biogenic particles.

The BioTrak instrument remains to be further characterized through comparisons with other bioaerosol monitoring instruments, using well-characterized fluorescent particles and by comparison to other offline techniques. Atmospheric $N_{FBAP}$ depends a lot on local sources. Moreover, $N_{FBAP}$ discrepancies between different studies, see Table 1, can also be explained by differences between different types of instruments. To understand how the bioaerosol data collected here compare to other long-term studies would require thorough intercomparisons between the different types of instruments that have been employed in different studies. The UV-APS measures autofluorescence between 420–575 nm after excitation with a 355 nm laser, at a flowrate of 1 Lmin[-1], while the WIBS measures autofluorescence between 310-400, and 400-600 nm upon excitation with 280 nm and 370 nm xenon lamps, at a flowrate of 0.3 Lmin[-1]. The BioScout excite particles with a 405 nm laser and measures autofluorescence at >442 nm with an adjustable flowrate, with default value 2 Lmin[-1]. The BioScout has been suggested to detect ambient bioaerosols more efficient than the UV-APS (Saari et al., 2014). The BioTrak was developed for the purpose of detecting bioaerosols in pharmaceutical production and clean environments, thus, it operates at a higher



flowrate, 28 Lmin$^{-1}$, than other LIF instruments. To further understand the potential of using the BioTrak in future studies, the BioTrak needs to be compared to other new technologies for automatic bioaerosol monitoring. Lieberharr et al. (2021) presented a potential standardized validation method for assessment of counting efficiency and fluorescent measurement of bioaerosol instruments that could be used for validation of the BioTrak instrument (Lieberherr et al., 2021). To fully understand how the BioTrak data compares to other bioaerosol measurements techniques, BioTrak measurements need to be benchmarked against offline filter analyses.

### 3.1.2 Monthly and seasonal trends in total supermicron aerosol particles

Total aerosol particle number concentration ($N_{TAP}$) did not exhibit the same patterns as the $N_{FBAP}$ concentrations. The $N_{TAP}$ concentrations are shown on monthly and seasonal scales in Fig. 2B and 3B (Fig. S1 shows the full time-series with seasons indicated in the supplement). Compared to the $N_{FBAP}$ concentrations, the TAP concentrations were more homogeneous over the full campaign. No distinct peak of $N_{TAP}$ during warmer periods was observed. The highest $N_{TAP}$ were measured in September, October, February, and March. Lowest $N_{TAP}$ were observed in January 2021 and in December 2022. The $N_{TAP}$ varied more in the fall and the winter compared to the summer, as indicated by the 5$^{th}$ and 95$^{th}$ percentiles in Fig. 3B. No significant difference in $N_{TAP}$ was observed over seasons. The $N_{FBAP}$ / $N_{TAP}$ ratio had its highest values in June, July, August, and September. The average highest relative contribution of FBAP (0.006) was observed in July.

Overall, $N_{TAP}$ measured here, between 3.42 and 4.96 cm$^{-3}$ as seasonal averages, were higher than in most other similar studies where both $N_{FBAP}$ and $N_{TAP}$ were measured. For instance, Schumacher et al. (2013) reported average $N_{TAP}$ between 0.41 and 0.47 cm$^{-3}$ in Finland, and between 0.20 and 0.73 cm$^{-3}$ in USA measured with a UV-APS, A.E. Valsan et. al. (2016) reported mean $N_{TAP}$ between 0.96 and 2.66 cm$^{-3}$ on a monthly scale in India measured with a UV-APS and Toprak et. al. (2012/2013) reported mean $N_{TAP}$ between 0.47 and 0.69 cm$^{-3}$ in Germany measured with a WIBS-4. Comparisons suggests that our $N_{TAP}$ measurements were about ~2-10 times higher than the $N_{TAP}$ previously reported in combination with $N_{FBAP}$ measurements. Note that these differences also influence the difference in ratios between $N_{FBAP}$ and $N_{TAP}$. One factor that could possibly help explain these differences is that the BioTrak instrument operates at a higher sample flow (28 Lmin$^{-1}$) compared to other LIF-instruments used which commonly have sample flows ranging from 0.3 to 2 Lmin$^{-1}$. But, to understand these differences in more detail, further analysis to understand local and regional sources, including other aerosol measurements are needed. Moreover, the BioTrak particle counting efficiency (reported to be 100% for particles >0.75 µm) could be tested and the BioTrak should be run side-by-side with other supermicron particle counters, such as an aerodynamic particle sizer (APS, TSI) for inter-comparison of the supermicron particle counting. It can also be noted that compared to other bioaerosol instruments the BioTrak counts particles in rather large size fractions.



### 3.1.3 Diel patterns

Variations in the daily bioaerosol concentrations were studied by averaging hourly $N_{FBAP}$ abundance for each season (see Fig.
S3 and Fig. S4 in the supplement). Daytime $N_{FBAP}$ was not significantly different from $N_{FBAP}$ measured during the night and
the daily variations were numerically very small. On an hourly basis, daily $N_{FBAP}$ and $N_{FBAP}/N_{TAP}$ peaked in the afternoon or
evening, when relative humidity was relatively low and the temperature high, in winter, spring and summer. In the fall, $N_{FBAP}$
peaked later in the evening.

Overall, the daily relative humidity curve was smooth and repeated the same pattern in all seasons: relative humidity was low
in daytime and increasing in the night due to decrease in temperature. In summer, the minimum relative humidity was aligned
with the $N_{FBAP}$ peak. In fall and spring however, the $N_{FBAP}$ peak was preceded by the minimum relative humidity. Wind speed
maximum was on average aligned with the daily temperature peak and average wind directions were on average coming from
the South in the morning and more from the South-West later in during the day, the evening, and the night.

### 3.1.4 Size of fluorescent biological aerosol particles

Over the full campaign the smallest particles, 1-3 µm, on average made up ~70 % of the total $N_{FBAP}$. As comparison, particles
with sizes 3-5 µm and 5-12 µm contributed ~25 % and ~5 %, respectively. The contributions from larger particles, 3-12 µm,
were higher in the spring and summer, compared to the fall and winter. The constantly elevated $N_{FBAP}$ in the 1-3 µm range was
consistent with previous observations (Artaxo and Hansson, 1995; Huffman et al., 2010; Schumacher et al., 2013; Healy et al.,
2014; Valsan et al., 2016). The seasonal difference also agreed with previously reported results (Schumacher et al., 2013;
Huffman et al., 2010). Figure 4 displays the $N_{FBAP}$ particle size distribution for all seasons divided into the 3 different size bins.


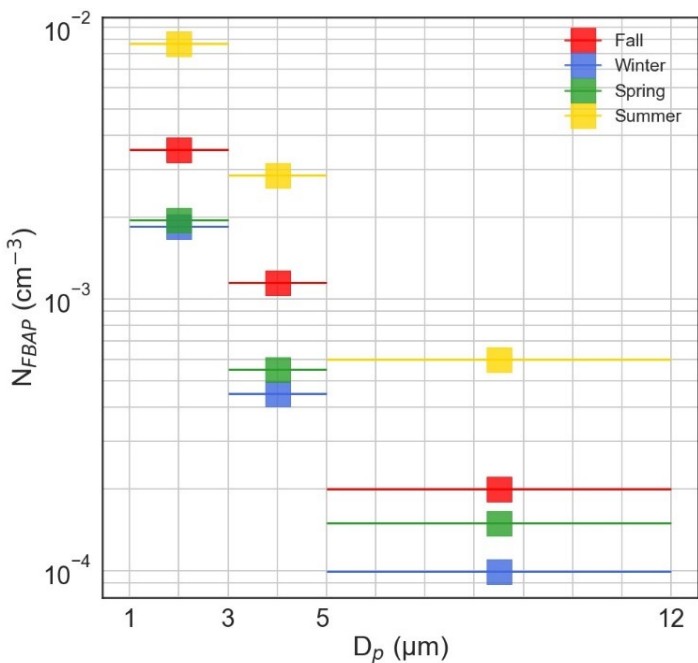


**Figure 4: Fluorescence biological aerosol particle (FBAP) number size distribution for the full campaign for three different size bins: 1-3µm, 3-5µm and 5-12µm.**

While LIF-measurements do not distinguish between different bioaerosol types, particle size can give some indication of the kind of particle. Single bacterial and fungal cells and fungal spores typically have sizes of 1-3 µm. Particles with sizes 3-5 and 5-12 µm correspond to the size of larger fungal spores, and smaller-sized pollen grains, although most pollen are too large to be sampled by the instrumentation used. The higher contribution of larger particles during spring and summer can potentially be explained by the spread of pollen during these seasons. Further confirmation and identification of the origin and the sizes

of the bioaerosols measured at Hyltemossa will be performed in follow-up studies with microscopic analysis but are beyond the scope of this work.

### 3.2 Meteorological effects

The meteorological conditions can have a variety of effects on the release and generation of biological aerosol particles into the atmosphere. In the following section, we investigate possible associations between $N_{FBAP}$ and air temperature, relative

humidity, wind effects and precipitation. It should be noted that we only consider the local and rapid changes in the parameters measured while for instance, atmospheric circulation has not been considered. Delays between changes in parameters and for instance changes in particle concentrations have also not been considered.



### 3.2.1 Air temperature effects

Figure 5 shows the observed relationship between $N_{FBAP}$ and temperature: median $N_{FBAP}$ binned based on air temperature

separately for each season (Fig. 5a) and the 7-day Pearson's rolling correlation coefficient $r$ for $N_{FBAP}$ with air temperature for the full campaign Fig. 5b). From the air-temperature-binned data $N_{FBAP}$ was significantly positively correlated with increasing ambient air temperature in spring (r=0.88, P<0.01; $\rho$ =0.83, P<0.01) and summer (r=0.88, P<0.0001; $\rho$ =0.94, P<0.0001). No significant associations between $N_{FBAP}$ and air temperature were identified in fall and winter. The 7-day rolling correlation coefficient $r$ based on hourly mean values shown in 5b indicate that for most of the campaign (60 % of the rolling correlation

periods studied) $N_{FBAP}$ increases were significantly (P<0.05) correlated with increasing air temperatures. It should be noted that the correlation coefficient for most part indicated only a moderate correlation ($r$ between 0.25 and 0.5) and the correlation was rarely strong ($r$ above 0.7). Negative correlations between $N_{FBAP}$ and air temperature were rarely observed (only in 13 % of the periods studied), but consistently only observed during winter months. The rolling correlation was nonsignificant during a significant part of the campaign (27 %). The presence of both positive, negative, and non-significant correlations between

$N_{FBAP}$ and temperature in the fall and the winter can explain why no overall consistent relationship between air temperature and $N_{FBAP}$ was observed. The data presented in Fig. 5 suggest that the processes that determine the release of FBAPs are strongly dependent on season and on mechanisms that potentially require a minimum temperature, or that are at least correlated with increasing air temperatures. The air temperatures ranged from -2 to 20 °C in the fall, from -10 to 10 °C in the winter, from -2 to 18 °C during spring, and from 6 to 30 °C in the summer.


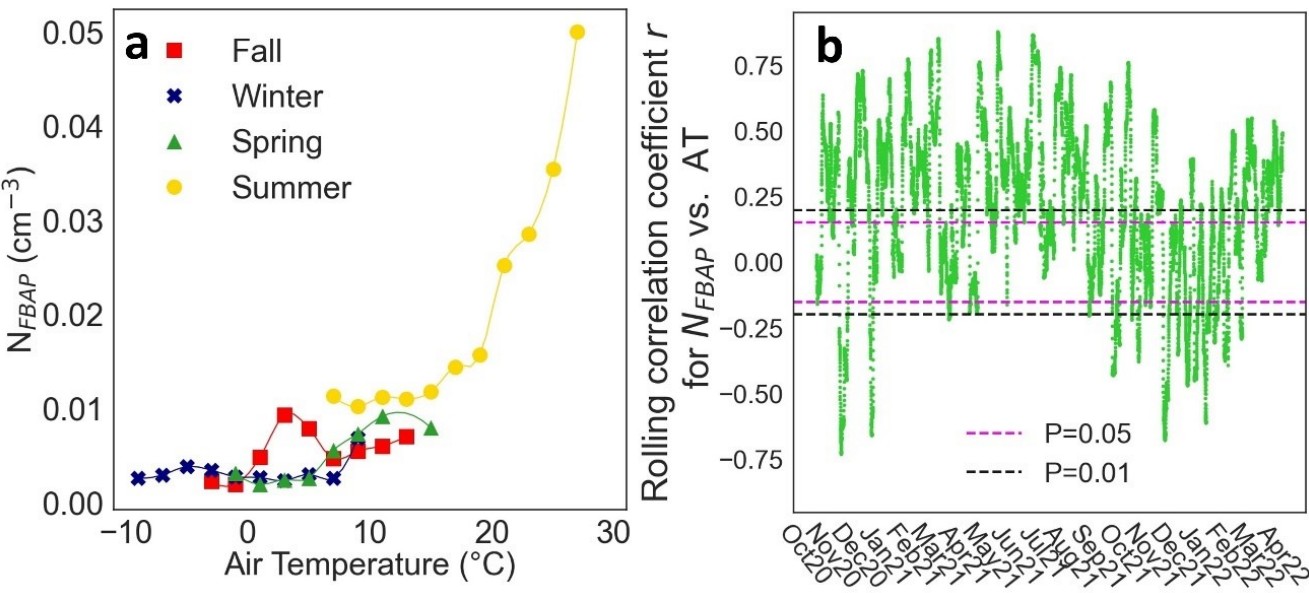

**Figure 5: Relationship between $N_{FBAP}$ and air temperature (AT). Median seasonal relationship between $N_{FBAP}$ and air temperature (a). Data was averaged into 21 bins between -10 and 31 °C. Bins that contained less than 0.1 % of the total data points were removed.**





**Fitted curves are included to guide the eye. Both N$_{FBAP}$ and air temperature variations were highest in the summer, as is also suggested in Figure 2 and 3. The 7-day rolling correlation coefficient *r* for hourly mean N$_{FBAP}$ with air temperature for the full campaign. Horizontal dotted lines indicate the range at which the correlation was nonsignificant at levels 0.05 (magenta) and 0.01 (black). In most cases a significant and positive correlation was observed between N$_{FBAP}$ and air temperature.**

### 3.2.2 Relative humidity effects

The observed relationship between N$_{FBAP}$ and relative humidity was complex and calls for more detailed studies. Figure 6 displays the average relationship between N$_{FBAP}$ and relative humidity: with N$_{FBAP}$ binned based on relative humidity (6a) and the 7-day rolling correlation coefficient for N$_{FBAP}$ and relative humidity (6b). On a seasonal level, N$_{FBAP}$ was positively correlated with relative humidity (r=0.21, P<0.05; $\rho$ =0.26, P<0.05) in the fall but negatively correlated with relative humidity

in summer (r=-0.65, P<0.05; $\rho$ =-0.71, P<0.0001) and in the winter (r=-0.14, P<0.05; $\rho$ =-0.12; P<0.05), while no correlation was found between relative humidity and N$_{FBAP}$ in the spring. The observed positive correlation between N$_{FBAP}$ and relative humidity in the fall may indicate that the bioaerosols detected in the fall were potentially generated or ejected due to relative humidity-dependent mechanisms. On the other hand, the negative correlation between N$_{FBAP}$ and relative humidity in winter and summer suggests that the high relative air humidity may be a limiting factor for the release of bioaerosols during those

seasons. In particular, the high N$_{FBAP}$ at relative humidity ~20-40 % suggests that relatively dry conditions increase the concentration of bioaerosols generated in the summer.

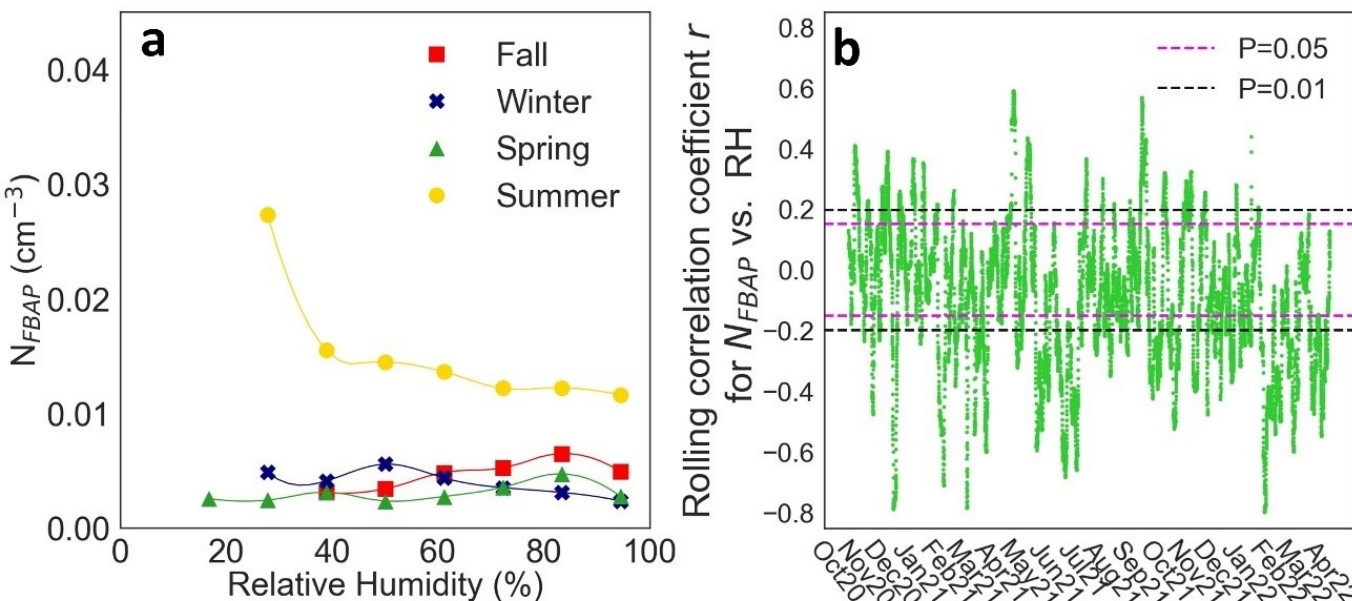

**Figure 6: Seasonal association between N$_{FBAP}$ concentrations and relative humidity. Median N$_{FBAP}$ concentration as a function of relative humidity for each season (a) and the 7-day rolling correlation coefficient *r* for hourly mean N$_{FBAP}$ and relative humidity (RH) for the full campaign (b). Horizontal dotted lines indicate the range at which the correlation was nonsignificant at levels 0.05 (magenta) and 0.01 (black). In most cases (46 %) the relationship observed was nonsignificant with very low correlation coefficients.**



From the 7-day rolling correlation coefficient $r$ between $N_{FBAP}$ and relative humidity for the full campaign, one can see that the correlation between $N_{FBAP}$ and relative humidity was mostly nonsignificant (46 %), while it was negative and significant for a large part of the campaign (39 %) and only rarely positive (14 %). The observed relationship that $N_{FBAP}$ and relative humidity are uncorrelated or negatively correlated are in contrast with what other studies report. In other long-term studies the connection between $N_{FBAP}$ and relative humidity have been reported as overall positive (Huffman et al., 2010; Huffman

et al., 2012; Schumacher et al., 2013; Toprak and Schnaiter, 2013; Valsan et al., 2016), although in some of the cases only low correlation coefficients (but not reported how low) were identified (Huffman et al., 2010), in other cases correlation coefficients and significance levels were not reported (Huffman et al., 2012) and the $N_{FBAP}$ and relative humidity relationship was inconsistent (Schumacher et al., 2013).

Relative humidity correlates inversely with air temperature since the temperature affects the saturation water vapor pressure in air. This relationship was also observed for the relative humidity and air temperatures measured ($r=-0.30$, $P<0.0001$; $\rho =-0.28$, $P<0.0001$), and the relationship can be observed in the daily trends in Fig. S3. Based on the positive relationship $N_{FBAP}$ had been observed to have with increasing air temperatures, it might be plausible to think that part of the negative correlation observed between relative humidity and $N_{FBAP}$ can be related to the relationship between relative humidity and air temperature.

However, it should be noted that the relationships are complex and that the causality between the different effects can only be interpreted after more detailed studies. The data here reported underlines the need for further studies of the relationship between relative humidity and $N_{FBAP}$ under different conditions and in different types of environments.

### 3.2.3 Wind effects

The $N_{FBAP}$ was positively correlated with increasing wind speed. Figure 7 displays the average relationship between $N_{FBAP}$ and wind speed for the four seasons (a) and for most of the campaign as assessed with the 7-day rolling correlation between $N_{FBAP}$ and wind speed(b). The $N_{FBAP}$ was positively significantly correlated with wind speed in the winter ($r=0.89$, $P<0.0001$; $\rho =0.95$, $P<0.0001$), spring ($r=0.70$, $P<0.05$) and summer ($r=0.64$, $P=0.05$; $\rho =0.76$, $P<0.05$). In the fall a similar but nonsignificant correlation was observed ($r=0.58$, $P=0.08$; $\rho =0.46$, $P=0.18$). These correlations were also confirmed by the

rolling correlation coefficient. Figure 7b shows that for a large part of the campaign time the relationship between $N_{FBAP}$ and wind speed was nonsignificant (42 %) or positive (39 %) and only rarely there was a negative and significant relationship (15 %).





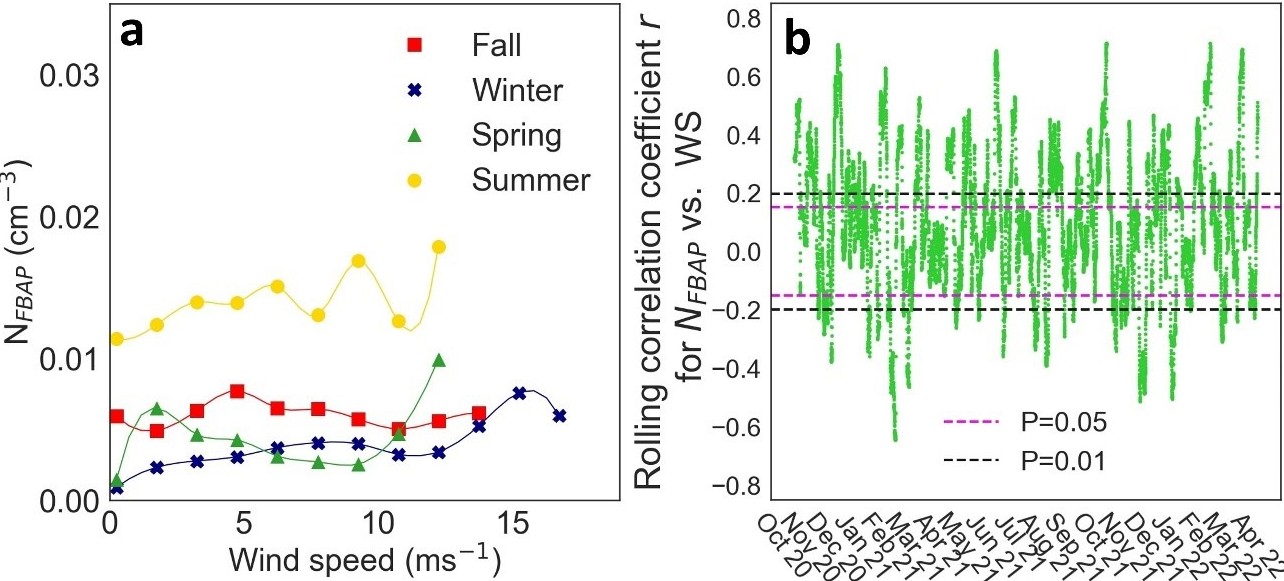

**Figure 7: The relationship between FBAP number concentrations and wind speed on a seasonal basis (b) and the rolling correlation for the relationship over the full campaign (b). An overall positive relationship was observed between NFBAP and wind speed for all seasons, which can be observed in both figures, but in many cases the relationship was nonsignificant.**

Figure 8 shows $N_{FBAP}$ as a function of wind direction. For all seasons, the winds coming from North-East to South-East correlated with the highest $N_{FBAP}$ abundances. Meanwhile, winds from South-West to North-East were in general associated with the lowest $N_{FBAP}$. It was also noted in the data that in fall and winter the coldest air temperatures were correlated with winds from the North, while air temperatures in the spring and summer were more independent of wind directions. Studies on long-range transport of air masses was beyond the scope of this study but could possibly have had been indicative for understanding these data fully.





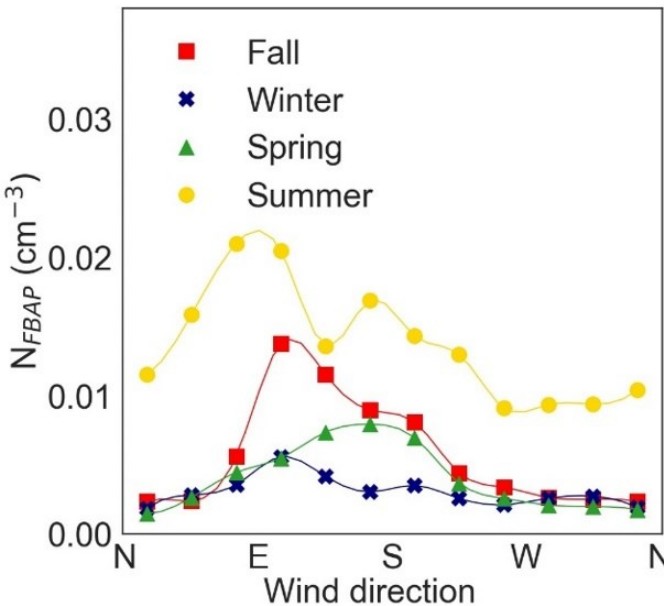

**Figure 8: The effect of wind direction on N$_{FBAP}$ abundance on a seasonal basis. The highest N$_{FBAP}$ concentrations were measured for winds from the East. The lowest N$_{FBAP}$ were measured when the wind blew from North to West.**

### 3.2.4 Precipitation effects

In some cases of rain, but not consistently for all rain events, a substantial increase of FBAP number concentrations were observed before, during and right after rain. For certain rain events, the FBAP concentration was observed to increase by factors 4-10 when compared to N$_{FBAP}$ outside of the rain event. The pattern was seen for all seasons. A total number of 90, 44, 35 and 64 individual rain events were identified in the fall, winter, spring, and summer, respectively. In about 50 % of all cases of a rain event, an immediate, but not lasting, increase in N$_{FBAP}$ concentrations was observed, but the effect was not statistically significant. The N$_{FBAP}$ concentrations varied a lot both before and after rain events, and sharp increases in N$_{FBAP}$ were also observed unrelated to rain. Overall, the FBAP concentration variations were large and therefore, sudden, and instantaneous variations, that could plausibly be explained by rainfall, were not significant over full seasons. Figure S5 and S6 in the supplement shows exemplary rain events and the distribution of N$_{FBAP}$ before, during, after rain and when there is no rain, respectively.

Only local rain was accounted for and therefore, the effect of rain upwind and possible transport of FBAP to the measurement site was not detangled. On average, a rain event lasted a few hours but during certain periods the frequency of such rain events was high. It should be noted that the overall number of rain events, and the increased relative humidity associated with such

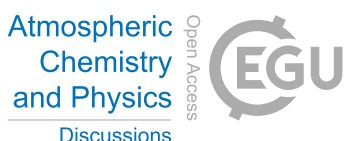

events, could have a larger and longer-ranging effect than here distinguished. The relationship between biological aerosol particles and rain have been reported on for a long time (Gregory and Hirst, 1957; Hirst and Stedman, 1963) and calls for standardized methods for the association between PBAP and precipitation. Rainfall can be important both for scavenging of bioaerosols as well as for the bioaerosol release.


### 3.2.5 Connecting fluorescent biological aerosol particle release mechanisms with meteorological effects

Connecting the observed results in this study with mechanisms for aerosolization and release of biological particles allows a greater understanding of seasonal variations in $N_{FBAP}$. In summer, there was a positive correlation with air temperature and wind speed, while the relationship was mostly nonsignificant or negative with relative humidity. This suggests that wind-

induced bioaerosol generation was favored during dry and warm conditions. This was likely the case during spring as well, when $N_{FBAP}$ correlated positively with air temperature and wind speed.

During fall, the highest number of rain events was observed, as well as a small positive correlation between $N_{FBAP}$ and relative humidity. Bioaerosols can be generated and dispersed by mechanical ejections and bubble bursting processes when raindrops

impact on surfaces (Alsved et al., 2019; Kim et al., 2019), which could explain the, in some cases strong, association between $N_{FBAP}$ and rain events. But rainfalls are also known to clean the air from aerosol particles in the lower troposphere (Moore et al., 2020), which can explain the inconsistent association between $N_{FBAP}$ and rain events. Although many studies have found a generally negative correlation between airborne spore concentrations and relative humidity, some fungal species absorb water from the air, causing swelling of the mucilage and subsequent explosive release of spores (Grinn-Gofron and Bosiacka, 2015).

High relative humidity is also known to cause pollen to rupture, resulting in the release of smaller pollen fragments (Taylor et al., 2004). It is also noteworthy that different types of bioaerosols can have different relationships with humidity and air temperature so that the common effects are masked. As noted by Oliviera et al. (2009) while some spore types have been observed to be negative correlations with temperature and positive correlation with relative humidity and rainfall, other spores showed the opposite correlations (Oliveira et al., 2009).


The lowest levels and variation of $N_{FBAP}$ were found during winter (Fig. 3a), likely due to the lower temperatures, sun light and biological activity during this season. Again, low relative humidity and wind speed were correlated with higher $N_{FBAP}$, indicating wind-induced aerosolization. In Southern Sweden, cold temperatures are often correlated with northern winds, which was observed for fall and winter in the meteorological data here studied. This agrees well with the lowest $N_{FBAP}$ levels

being found when winds came from the north. It should also be noted that the difference in wind direction could be indicative of different sources of bioaerosols.



For further understanding of the data, FBAP release mechanisms and FBAP sources, detailed biological analyses, including fluorescence microscopy of PBAP filter sampling, are needed to identify the different types of bioaerosols that are present
during different seasons.

**4 Conclusions**

Fluorescent bioaerosols were measured continuously in real-time during 18 months in the Southern Sweden. To our knowledge, this is the first report of this instrument being operated for outdoor measurements and for such a long period of time. Large variations in both fluorescent biological aerosol particle concentration and supermicron particle number
concentrations were observed. Over the full measurement period, the average $N_{FBAP}$ concentration was 0.005 cm$^{-3}$ and the monthly average varied between 0.001 cm$^{-3}$ and 0.020 cm$^{-3}$. The $N_{FBAP}$ concentration was highest in the summer (median 0.02 cm$^{-3}$) and lowest in the winter (median 0.004 cm$^{-3}$).

Total aerosol particle concentrations did not follow the same trends as the $N_{FBAP}$ concentrations. Instead, $N_{TAP}$ remained
relatively constant throughout the year. These data indicate that the sources of fluorescent bioaerosols were not the same as for the non-fluorescent particles. Further, this suggests that the $N_{FBAP}$ concentrations were more influenced by the biological activity than by boundary layer meteorology. No differences were found between daytime and nighttime $N_{FBAP}$ concentrations and overall daily variations were minimal compared to variations over seasons. The 1-3 μm $N_{FBAP}$ particles made up on average 70 % of the total $N_{FBAP}$ abundance, which suggest that the largest number contribution to PBAP was the occurrence of single
bacterial and fungal cells, fungal spores, and agglomerated bacteria.

Overall, these long-term measurements confirm that the emission and abundance of biological aerosol particles in rural environments were closely related to meteorological parameters. Over the full campaign $N_{FBAP}$ was positively correlated with air temperature (P<0.01) and wind speed (P<0.01), while the relationship was more complex, and more negatively correlated,
between $N_{FBAP}$ and relative humidity (P<0.05). No significant relationship was observed between rain events and $N_{FBAP}$ over seasons, however several rain events gave rise to immediate and strong increases in $N_{FBAP}$ over short time periods. Our measurements indicate that bioaerosols were emitted due to mechanical wind release during the warmer seasons but also suggest that bioaerosol generation increased during wet conditions and increasing relative humidity in the fall. It is plausible that the balancing of two effects is present when the environment is wet: scavenging of bioaerosols due to rainfall and
generation and release due to rainfall. Data here presented also suggest that biological aerosol release was prohibited in the winter.



Long-term data on biological aerosol particles are lacking from the North of Europe but also from all over the globe. This study presents a first attempt to analyze and understand 18 months of data on atmospheric fluorescent biological aerosol particles measured with a LIF-instrument.



**Code availability**.

The code used to produce the results of this study is available from the first author (MPS) upon qualified request.

**Data availability.**

The meteorological data is publicly available from the carbon portal (https://www.icos-cp.eu/observations/carbon-portal ) with

pid: 11676/tAq_SRIWDxBoVJp7_klS8ZbA. The data are available here:

https://hdl.handle.net/11676/jW7oCGwqLrA4JsrPH5dh78On (Heliasz, 2022b) and here:

https://hdl.handle.net/11676/L27iDe53nai2M5MSKRaZM6Jo (Heliasz, 2022a).

**Author contributions.**

MPS was responsible for data taking, performed the analysis, produced the figures and wrote the initial draft of the manuscript.

MA significantly contributed with guidance, analysis input and writing the manuscript. TST contributed with expertise and input on analysis and the manuscript. TBK significantly contributed to the performed analysis, advised on the analysis and contributed significantly to the manuscript writing. JL was overall responsible for the study, advised on the analysis and contributed significantly to the manuscript.

**Competing interests.**

The authors declare that they have no conflict of interest.

**Acknowledgements.**

This work was supported by the Swedish research council for sustainable development FORMAS (grant numbers 2017-00383 and 2020-

01490) and AFA insurance (grant numbers 180113 and 200109). The authors would like to acknowledge the Aerosols, Clouds, and Trace gases Research InfraStructure (ACTRIS) for letting us put up the instrument at their site, and the integrated carbon observation system (ICOS) for providing meteorological data for this study. We thank and acknowledge Patrik Nilsson, Design Sciences, Lund University, for helping with the instrument installation, Jonas Jakobsson, Erik Ahlberg, Adam Kristensson, Department of Physics, Lund University, and Marcin Jackowicz-Korczynski, Department of Physical Geography, Lund

University, for helping with data taking at Hyltemossa. We thank Erik Swietlicki for input and comments on the manuscript. We acknowledge ICOS Sweden for provisioning of data (pid: 11676/tAq_SRIWDxBoVJp7_klS8ZbA) and we would like to thank Tobias Biermann and Michal Heliasz at the Centre for Environmental and Climate Science (CEC) and ICOS Sweden for assistance. ICOS Sweden is funded by the Swedish Research Council as a national research infrastructure.




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
