# Peer review of "Measurement report: Atmospheric fluorescent bioaerosol concentrations measured during 18 months in a coniferous forest in the south of Sweden"

_Atmospheric Chemistry and Physics, 2022_

## Referee Comment (RC2)

This study reports the number concentration of fluorescing particles as well as the total supermicron (1-12 µm) aerosol particle number concentration at a rural site in Sweden over a period of 18 months. The authors employed the BioTrak monitor, which provides number concentration of viable particles as well as total particle number concentration in real time. Real-time bioaerosol monitoring is highly relevant for allergy/asthma prevention, for ecology (e.g. for monitoring invasive plant species) and for studying the effects of climate change (e.g. the growth/spread of vegetation towards higher altitudes or in the arctic region). This manuscript provides useful insight into the release of biological aerosol particles and their relationship to different meteorological parameters and falls well within the scope of the journal Atmospheric Chemistry and Physics. My main concern (see comments below) is that the performance characteristics of the BioTrak monitor are not known. Without any knowledge on the counting efficiency of the OPC module and the sensitivity of the LIF detector, it is impossible to assess how quantitative the results are. It is also impossible to compare these data to data published in previous studies using different monitors. Although this issue is certainly not specific to this study – instrument calibration in the supermicrometre particle range is for various reasons much more challenging than in the submicrometre range – it is still troubling.

In my opinion, the authors should provide some evidence on the performance of the BioTrak before this manuscript is accepted for publication.

Comments/questions

1) Page 2/Line 57: The authors argue that "Progress in the detection of bioaerosols with higher time resolution has been made by utilizing laser-induced fluorescence (LIF) (Hill et al., 1995)". Apart from LIF, new hybrid instruments have become available in the last 4-5 years, which combine different detection methods with machine learning. Such instruments include the Poleno (Swisens, Switzerland), the Rapid-E (Plair, Switzerland) and the BAA 300 (BAA 500, Hund GmbH)

Sauvageat et al. 2019 https://doi.org/10.5194/amt-2019-427

Šaulienė et al. 2019 https://doi.org/10.5194/amt-12-3435-2019

J. Schiele *et al.*, "Automated Classification of Airborne Pollen using Neural Networks," *2019 41st Annual International Conference of the IEEE Engineering in Medicine and Biology Society (EMBC)*, 2019, pp. 4474-4478, doi: 10.1109/EMBC.2019.8856910.

These hybrid methods have shown very promising results in real-time bioaerosol identification and counting and seem to be more powerful than LIF alone.

2) Text: Please insert a space between L and $min^{-1}$ ($Lmin^{-1}$ -> $L\ min^{-1}$). Please insert a space before the unit $^{o}C$ (e.g. $10^{o}C$ -> $10\ ^{o}C$). Similarly, $ms^{-1}$ -> $m\ s^{-1}$, $gcm^{-3}$ -> $g\ cm^{-3}$, $mmh^{-1}$ -> $mm\ h^{-1}$

3) Page 12/ Line 263: The name of the author is Lieberherr et al. (2021)

4) Page 12/Line 289: The authors argue that "the BioTrak should be run side-by-side with other supermicron particle counters, such as an aerodynamic particle sizer (APS, TSI) for inter-comparison of the supermicron particle counting". According to a recent study (Vasilatou et al. 2022 https://doi.org/10.1080/02786826.2022.2139659), APS units exhibit a significant unit-to-unit variability and the counting efficiency drops fast at particle sizes ≥10 µm. Moreover, it is not straightforward to compare the BioTrak, which is an optical particle counter, with an aerodynamic particle counter since these instruments classify particle size differently. It might make more sense to run the BioTrak side-by-side with other optical particle size spectrometers, e.g. the OPS 330 (TSI Inc, USA) or different OPSS by Grimm GmbH, Germany. OPSS tend to be more robust, therefore more suitable for field studies, and have been characterised more extensively in the laboratory.

5) Page 12: The authors state that "To further understand the potential of using the BioTrak in future studies, the BioTrak needs to be compared to other new technologies for automatic bioaerosol monitoring. Lieberherr et al. (2021) presented a potential standardized validation method for assessment of counting efficiency and fluorescent measurement of bioaerosol instruments that could be used for validation of the BioTrak instrument (Lieberherr et al., 2021)." I would like to note that there is a second method available for calibrating optical particle counters based on the Inkjet Aerosol Generator (Iida et al. 2013 https://doi.org/10.1063/1.4803302). This method could also be applied to bioaerosol monitors.

6) Considering that there exist at least two methods for calibrating optical particle counters, I am wondering why the authors did not attempt to calibrate their instrument before or after their field campaigns. I realise that the calibration of the LIF detector, which determines particle viability, is not standardised and that certified fluorescent reference materials are not commercially available. However, the calibration of the first detector/counter measuring scattered light should be possible with standardised procedures laid out in ISO 21501-4. Without any prior characterisation of the BioTrak, the results of this study remain in the best case semi-quantitative. That would be a pity, considering the great efforts and hard work that the authors invested in this lengthy field campaign.

I strongly recommend that the authors provide some evidence on the performance (especially the counting efficiency) of the BioTrak. If it is not possible to send the instrument to a metrology institute or any other accredited laboratory for calibration, I suggest that you compare the BioTrak with a calibrated OPSS at your premises or in the field. In this case, the OPSS would be used as a transfer standard. Providing more information on the performance of the BioTrak will enhance the quality and impact of this otherwise very meticulous work.

Minor corrections and suggestions for improvement

Page 3/Line 77: Consider revising this sentence as follows "In the studies  listed here…"

Page 3/Line 81: Please insert a space between the number 10 and the unit µm

Page 12/Line 286: Consider revising this sentence as follows "However, to understand these differences in more detail…".

Page 18 / Line 418: Consider revising this sentence as follows "Studies on long-range transport of air masses  were beyond the scope of this study but could have possibly contributed to  a better understanding of these data .

Page 19 / Lines 426-428: Consider revising this sentence as follows "… a substantial increase of FBAP number concentrations was  observed before, during and right after rain… , the FBAP concentration was observed to increase by a factor of 4-10).

Page 21 / Line 478: Consider amending this sentence as follows: "Fluorescent bioaerosols were measured continuously in real-time during 18 months in the Southern Sweden using a BioTrak".

Page 21 / Line 500: Consider revising this sentence as follows "The data  presented in our study…".

---

## Author Comment (AC1)

We thank the reviewers for their careful reading of the manuscript. In the following we address all the comments and remarks in the order we received them.

*Answers to RC1:*

General remarks

Although, I don't have serious doubts that the applied instrument (BioTrak) is able to measure biological aerosol particles in ambient air, the manuscript could be much stronger if the authors would include a discussion on the known quality of the instrument. As the instrument was not applied yet to ambient air, probably some conclusions on its validity could be drawn from other applications. I'd be surprised if there were no reports on validation experiments in the literature or by the manufacturer. Furthermore, in section 2.2 a comparison of the BioTrak and instruments used for previous studies (UV-APS, WIBS), including, if possible, a discussion of potential implications, would be very helpful for the reader."

**Response: Thank you for underlining this. We have now included a discussion on the validity of using the instrument for ambient air. We have addressed this in our new version of the manuscript, see section 3.3 in the Discussion. We have performed a validation of the OPC in the BioTrak with a Grimm OPC. This is discussed in section 3.3 and also complemented by new figures in the supplement (Figures S7 and S8). We have also included a comparison between BioTrak and UV-APS and WIBS in lines 140-148 in section 2.2**

Specific comments

l. 21: ' a ~3-fold difference between these seasons'. Which values does this statement relate to? Dividing the summer (0.0126 cm-3) by the winter value (0.0025 cm-3) results in a factor of 5.

**Response: Thank you for noticing this error. We have updated the new version with the correct number. See line 21.**

l. 134: Was there any evaluation of the BioTrak instrument against different methods, e.g. by this or previous studies? If not for ambient aerosol, then maybe in pharmaceutical and clean environments, which is mentioned later in the manuscript as the purpose of the BioTrak instrument.

**Response: We have added a discussion on the validity of using the instruments for ambient air measurements in section 3.3. In the new version of the manuscript, we also refer to the BioTrak manufacturer's validation of the instrument. To the best of our knowledge, this is the only validation that exists so far.**

l. 146-149: Since the N_FBAP was measured within the canopy (5m), it would be interesting to know or estimate how in particular air temperature and humidity deviate between 30m / 70m and within canopy values.

**Response: On average the temperature was 1.8 degrees Celsius higher at 30 m compared to at 70 m height. Relative humidity was on average 4 % higher at 30 m compared to 70 m. We also address this now on lines 161-164 in the manuscript.**

l. 225-226: Was this also observed during this study? The sentence could be understood in this way and that this compares with other studies.

**Response: Yes, we believe this is also the reason to the increase in measured bioaerosol concentrations. We have clarified this in lines 240-243.**

l. 282-284: Since Hyltemossa is an ACTRIS and ICOS site, are there other aerosol number concentration or size distribution measurements available or at least nearby?

**Response: There are other measurements available, but these are only of fine particles. Therefore, no comparisons have been made with other aerosol measurements at the site. In the future, when more micron sized particle measurements are planned to take place at the site, we will compare these to the Biotrak measurements. We addressed this in the reviewed version of the manuscript, see lines 309-313.**

l. 284-286: Could the authors please explain in a few more sentence how the higher flow rate might influence the N_TAP concentration? In case the higher flow rate would have an influence on the N_TAP concentration, how could it affect the N_FBAP concentration?

**Response: The comparison performed between the BioTrak and the Grimm OPC indicated that the BioTrak correctly estimates the total aerosol particle concentration measured at Hyltemossa. It is therefore assumed that the major differences between this long-term campaign and other long-term bioaerosol and TAP measurements are mainly due to the different environments. See line 304-306 in the new manuscript.**

l. 288: How does the counting efficiency and cut off size compare to the other instruments (UV-APS, WIBS)? Presumably, a difference of a few hundred nm in cut off size in the accumulation mode might result in large differences in N_FBAP (and perhaps N_TAP if done also within the UV-APS / WIBS instruments and not externally in other studies).

**Response: Yes, this is of course also an important issue. The other instruments measure particles down to 0.5 and 0.3 µm in diameters. We have clarified this in section 2.2**

section 3.2.2: Can the relative humidity affect the measurement signal of the BioTrak?

**Response: The aerosol particles were not dried prior to BioTrak air sampling, which could cause growth of particles due to increased relative humidity and hence particles to be wrongly classified as larger sized particles. We have addressed this in section 3.2.2 line 390-392.**

section 3.2.3: Are the wind direction results significant? If yes, wouldn't this indicate a measurable difference between, roughly speaking, north-westerly and south-easterly wind directions in terms of FBAP sources at the Hyltemossa site?

**Response: The differences were not significant. We mention this in the manuscript now on lines 440-443.**

section 3.2.4: As mentioned in the figure caption to fig. S5, there is some indication in fig. S6a for increasing N_FBAP with increasing rain intensity. Could you explain more, what in Fig. S6a is indicating this?

**Response: Yes, we have added information to this interpretation on lines 460-462.**

How do the statistical results change if different higher maximum rain intensity thresholds within a rain event would be used (not just on hourly basis but also shorter time periods)? Does a higher threshold such as 1 mm hr-1 or higher lead to a more robust, maybe even significant correlation

between N_FBAP an intense rain? However, this might limit the number of rain events in the statistics per season.

**Response: As you suggested, we also tested this, but the correlation between NFBAP and precipitation was still nog significant. We also mention this in the discussion at lines 451-453.**

l. 470-471: Such could be elaborated a bit more in section 3.2.3.

**Response: We have addressed it in lines 440-442.**

section 4: The size range (1-12 μm) in which FBAP could be observed with the BioTrak instrument should be mentioned in this summarizing section.

**Response: Thank you for noticing this. We have added the information on line 517.**

l. 481-482: The summer and winter median values differ from those given in the abstract.

**Response: Thank you for noticing this, we have updated the values given in lines 521-523.**

Also there is little discussion of figure 3 in the respective section.

**Response: We agree, we have added conclusions drawn based on Figure 3 in lines 522-524.**

l. 486-487: 'this suggests that the NFBAP concentrations were more influenced by the biological activity than by boundary layer meteorology.'? Could you please elaborate a bit more on the basis for this conclusion?

**Response: We have revised this sentences and instead written: "It can also be assumed that local meteorology affected the sources in different ways. " see line 528.**

Other minor comments, typos, etc.

l. 77: here listed -> listed here

**Response: Thank you, we changed it to "in the studies listed in Table 1" to clarify even more. See line 83.**

l. 78-79: Perhaps mention in the text as well that these values are averages over the time period of different campaigns. I think, giving also the maximum observed concentrations (if they have been reported) would be interesting for potential readers. Furthermore, the lowest value given in the table is 0.0053 cm-3, not 0.015 cm-3.

**Response: We now clarify this in line 82-84. We also updated the number as we had wrongly cited the wrong one in table 1, see line 84-85.**

l. 80-81: The authors cite here only 2 of the eight studies, but the table mentions a number fraction for most of the studies. Further, the minimum reported number fraction is 1.1 % (only season mean though).

**Response: You are correct the minimum number fraction should be 1.1 %, we have updated this in the manuscript on line 88. We have also clarified that we wanted to say that so far only two studies were long enough to address seasonal variations. We clarified this on line 86-88.**

l. 154-155: Since the authors have also used hourly data (e.g. for the diurnal cylce), the statement on averages over days, months and seasons is misleading. Perhaps the first sentence of this paragraph is not really needed.

**Response: We agree with you and have now removed this line.**

Table 2 and throughout the report: Since the ratio N_FBAP/N_TAP is so small, using % leads to easier understandable (shorter) numbers (as was done in the abstract). Also, the author's could consider to provide values of the N_FBAP in scientific notation instead of decimals for easier readability.

**Response: We agree that the readability could be improved by your suggestion, however we have chosen to keep the applied format since this is in-line with the already existing literature on the subject and therefore makes it easier to compare the results**.

l. 275-276: Suggestion to add the year of the given months.

**Response: We have corrected one faulty given month here, changing September to October, and also inserted the years. See lines 291-292**

l. 278: 'here' -> 'in this study'. Just sounds better in my view.

**Response: We updated this as suggested, see line 296**

l. 280: A.E. Valsan et al. -> Valsan et al.

**Response: Thank you, we have corrected this error. See line 298**

l. 281: 2012/2013 -> 2012, 2013

**Response: We have corrected this as suggested, see line 299**

l. 290-291: This might be very important. This statement could be mentioned earlier (e.g. section 2.2?) together with the difference in flow rate.

**Response: Thank you, we agree. We have added this to information in section 2.2**

l. 330: 'effects' seems redundant as 'associations' was mentioned in the beginning of this sentence.

**Response: Yes, we replaced effects with (magnitude and direction) see line 349**

l. 336: 'Fig. 5b)' misses a '('.

**Response: Thank you, we updated this. See line 356**

l. 339: 'fig.' missing before '5b'.

**Response: We added it, see line 359**

l. 339: indicate -> indicates

**Response: We have updated this, see line 359**

l. 343-344: Mentioning 'nonsignificant' and 'signifcant' in the same sentence, although for different things, is a bit distracting. Perhaps, 'significant' could be substituted (e.g., large, considerable, ...)?

**Response: We agree, we substituted significant for considerable, see line 364**

l. 348-349: If needed at all, this sentence should be moved to the beginning of section 3.2.1.

**Response: We agree we moved it to the beginning of section 3.2.1, see line 353**

l. 377 (caption of Figure 6): The horizontal lines are dashed not dotted.

**Response: Thank you, we updated this for captions in Figure 6 and Figure 5**

l. 396: here reported -> reported here

**Response: Thank you we changed this, see line 417.**

l. 402: Space missing between 'speed' and '(b)'

**Response: We changed this, see line 425**

l. 415: correlated -> are correlated. Sounds more familiar to me, but I am not a native English speaker.

**Response: We changed it to "were correlated" see line 437**

l. 463: 'to be negatively correlated' or 'to have negative correlations', similarly adapt 'positive correlations' later in that sentence.

**Response: Yes we updated these accordingly, see lines 490-491.**

l. 477: 'the' can be deleted in 'in the Southern Sweden'.

**Response: We updated this accordingly, see line 495**

figure caption of Figure S2: The 5min data averages are shown by a red line. The caption mentions 'small dots'.

**Response: We updated this to "The thin red line" see caption for Figure S2.**

figure caption Figure S3: ')' missing after 'time of the day'.

**Response: We updated this. See caption for Figure S3.**

*Answers to RC2:*

This study reports the number concentration of fluorescing particles as well as the total supermicron (1-12 μm) aerosol particle number concentration at a rural site in Sweden over a period of 18 months. The authors employed the BioTrak monitor, which provides number concentration of viable particles as well as total particle number concentration in real time. Real-time bioaerosol monitoring is highly relevant for allergy/asthma prevention, for ecology (e.g. for monitoring invasive plant species) and for studying the effects of climate change (e.g. the growth/spread of vegetation towards higher altitudes or in the arctic region). This manuscript provides useful insight into the release of biological aerosol particles and their relationship to different meteorological parameters and falls well within the scope of the journal Atmospheric Chemistry and Physics. My main concern (see comments below) is that the performance characteristics of the BioTrak monitor are not known. Without any knowledge on the counting efficiency of the OPC module and the sensitivity of the LIF detector, it is impossible to assess how quantitative the results are. It is also impossible to compare these data to data published in previous studies using different monitors. Although this issue is certainly not specific to this study – instrument calibration in the supermicrometre particle range is for various reasons much more challenging than in the submicrometre range – it is still troubling.

In my opinion, the authors should provide some evidence on the performance of the BioTrak before this manuscript is accepted for publication.

**Response: Thank you for this comment. We have added a validation of the BioTrak OPC with a Grimm OPC. See new section 3.3 "Using the BioTrak for ambient air measurements" on line 503 and Figures S8 and S9 in the Supplement. We have also addressed the differences between different types of bioaerosol monitors, see section 2.2 line 140-148 and forward.**

Comments/questions

1) Page 2/Line 57: The authors argue that "Progress in the detection of bioaerosols with higher time resolution has been made by utilizing laser-induced fluorescence (LIF) (Hill et al., 1995)". Apart from LIF, new hybrid instruments have become available in the last 4-5 years, which combine different detection methods with machine learning. Such instruments include the Poleno (Swisens, Switzerland), the Rapid-E (Plair, Switzerland) and the BAA 300 (BAA 500, Hund GmbH)
Sauvageat et al. 2019 https://doi.org/10.5194/amt-2019-427
Šaulienė et al. 2019 https://doi.org/10.5194/amt-12-3435-2019
(Schiele et al., 2019)*2019 41st Annual International Conference of the IEEE Engineering in Medicine and Biology Society (EMBC)*, 2019, pp. 4474-4478, doi: 10.1109/EMBC.2019.8856910.
These hybrid methods have shown very promising results in real-time bioaerosol identification and counting and seem to be more powerful than LIF alone.

**Response: Thank you for providing these references. We have looked into them and also referenced them appropriately in the section you mentioned. See lines 68-72.**

2) Text: Please insert a space between L and min-1 (Lmin-1 -> L min-1). Please insert a space before the unit ⁰C (e.g. 10⁰C -> 10 ⁰C). Similarly, ms-1 -> m s-1, gcm-3 -> g cm-3, mmh-1-> mm h-1

**Response: Thank you, we have done this now throught the manuscript. See for example section 2.3**

3) Page 12/ Line 263: The name of the author is Lieberherr et al. (2021)

**Response: Thank you for pointing this out. We have updated the name accordingly see line 280.**

4) Page 12/Line 289: The authors argue that "the BioTrak should be run side-by-side with other supermicron particle counters, such as an aerodynamic particle sizer (APS, TSI) for inter-comparison of the supermicron particle counting". According to a recent study (Vasilatou et al. 2022 https://doi.org/10.1080/02786826.2022.2139659), APS units exhibit a significant unit-to-unit variability and the counting efficiency drops fast at particle sizes ≥10 μm. Moreover, it is not straightforward to compare the BioTrak, which is an optical particle counter, with an aerodynamic particle counter since these instruments classify particle size differently. It might make more sense to run the BioTrak side-by-side with other optical particle size spectrometers, e.g. the OPS 330 (TSI Inc, USA) or different OPSS by Grimm GmbH, Germany. OPSS tend to be more robust, therefore more suitable for field studies, and have been characterised more extensively in the laboratory.

**Response: Thank you for pointing this out. We have now also run the BioTrak side-by-side with a Grimm OPC, which is a widely used instrument with a similar sizing principle. The results are showed in Figures S8 and S9 in the Supplement. We also address it in lines 504-515**

5) Page 12: The authors state that "To further understand the potential of using the BioTrak in future studies, the BioTrak needs to be compared to other new technologies for automatic bioaerosol monitoring. Lieberherr et al. (2021) presented a potential standardized validation method for assessment of counting efficiency and fluorescent measurement of bioaerosol instruments that could be used for validation of the BioTrak instrument (Lieberherr et al., 2021)." I would like to note that there is a second method available for calibrating optical particle counters based on the Inkjet Aerosol Generator (Iida et al. 2013 https://doi.org/10.1063/1.4803302). This method could also be applied to bioaerosol monitors.

**Response: Thank you for pointing this out. During our future validations of the BioTrak we will also consider this instrument. See lines 283-285.**

6) Considering that there exist at least two methods for calibrating optical particle counters, I am wondering why the authors did not attempt to calibrate their instrument before or after their field campaigns. I realise that the calibration of the LIF detector, which determines particle viability, is not standardised and that certified fluorescent reference materials are not commercially available. However, the calibration of the first detector/counter measuring scattered light should be possible with standardised procedures laid out in ISO 21501-4. Without any prior characterisation of the BioTrak, the results of this study remain in the best case semi-quantitative. That would be a pity, considering the great efforts and hard work that the authors invested in this lengthy field campaign.

**Response: Thank you for this comment and for seeing the potential of this study. Prior to the campaign the BioTrak was sent for calibration with Brookhaven Instruments in Sweden. We have now also performed validation of the optical particle counter in the BioTrak after the campaign. The results are presented in Figures S8 and S9 in the supplement and discussed in Sect 3.3, see lines 503-515.**

I strongly recommend that the authors provide some evidence on the performance (especially the counting efficiency) of the BioTrak. If it is not possible to send the instrument to a metrology institute or any other accredited laboratory for calibration, I suggest that you compare the BioTrak with a calibrated OPSS at your premises or in the field. In this case, the OPSS would be used as a transfer standard. Providing more information on the performance of the BioTrak will enhance the quality and impact of this otherwise very meticulous work.

**Response: Thank you for this comment and for seeing the potential of this study. We strongly agree with you and have therefore performed validation of the optical particle counter in the BioTrak. The results are presented in Figures S8 and S9 in the supplement and discussed in Sect 3.3, see lines 503-515.**

Minor corrections and suggestions for improvement
Page 3/Line 77: Consider revising this sentence as follows "In the studies here listed here…"

**Response: Thank you, we have updated the sentence to "In the studies listed in table 1..", se line 83-85.**

Page 3/Line 81: Please insert a space between the number 10 and the unit μm
**Response: We have now done this see line 88.**

Page 12/Line 286: Consider revising this sentence as follows "HoweverBut, to understand these differences in more detail…".
**Response: We changed it to "However,.." see line 304.**

Page 18 / Line 418: Consider revising this sentence as follows "Studies on long-range transport of air masses was were beyond the scope of this study but could have possibly contributed to have had been indicative for a better understanding of these data fully.

**Response: Thank you we have updated the sentence, see line 442-443**

Page 19 / Lines 426-428: Consider revising this sentence as follows "… a substantial increase of FBAP number concentrations was were observed before, during and right after rain… , the FBAP concentration was observed to increase by a factor of 4-10).

**Response: Thank you, we have updated the sentences accordingly. See line 449-450.**

Page 21 / Line 478: Consider amending this sentence as follows: "Fluorescent bioaerosols were measured continuously in real-time during 18 months in the Southern Sweden using a BioTrak".
Page 21 / Line 500: Consider revising this sentence as follows "The data here presented in our study…".

**Response: Thank you, we have updated the sentences to "Fluorescent bioaerosols, in the size range 1-12 μm, were measured continuously in real-time during 18 months in the Southern Sweden using a BioTrak" (lines 517-519) and "The data here presented in our study suggest that biological aerosol release was prohibited in the winter." (lines 543-544)**

---

## Author Response (AR2)

Dear Editor,

We would like to thank you for the opportunity to resubmit another revised copy of our manuscript *"Measurement report: Atmospheric fluorescent bioaerosol concentrations measured during 18 months in a coniferous forest in the south of Sweden"*.

The comments and remarks have helped us to further strengthen our paper. Please see below for a point-by-point response to all the referee comments.

Thank you in advance for your consideration, we look forward to hearing from you soon.

Best wishes,
Madeleine Petersson Sjögren and Jakob Löndahl (on behalf of all authors)

*Answers to referee #1 comments and remarks:*

Referee #1:
l. 140: ‚This is the first report on use of the BioTrak for measurement of bioaerosols.' According to l. 129 and suggested by its name, measuring bioaerols is a main purpose of the BioTrak. Probably the authors mean first application of the BioTrak for ambient air as stated in l. 504.

**Answer: You are correct. We updated the sentence to: "This is the first application of the BioTrak for measurement of ambient air."**

l. 141-142: The first sentence of this paragraph reads a bit out of context since both WIBS and UV-APS nor a comparison to measurements with these instruments were not mentioned in recent text passages.

**Answer: We agreed. We made the decision to introduce this paragraph in a slightly different way now. The following sentence was added: "For comparison of results between different studies of ambient air bioaerosol measurements (in particular with more used techniques such as UV-APS and WIBS), it should be noted that the BioTrak's fluorescence excitation and fluorescence emission operates at partly different wavelengths compared to both the WIBS and the UV-APS. Therefore, data are not completely comparable." See lines 141-144.**

l. 241-243: The sentence needs a grammatical revision. I don't understand what the authors wanted to express. Further, how could pollen from spring and summer survive until fall to increase the bioaerosol concentration in fall?

**Answer: We updated the two sentences to the following which is what we wanted to say: "At the beginning of fall spores from fungi are dispersed in the air, which increases the fall $N_{FBAP}$ (Schumacher et al., 2013; Toprak and Schnaiter, 2013). This also explains the pattern here observed, where an increase of bioaerosols in the fall is plausibly due to a combination of pollen dispersed from the late summer, in combination with fungi and spores in late summer and early fall (Sept-Oct, primarily)."**

section l. 296 - 314: Can the difference in cut off size between UV-APS, WIBS and Biotrak also influence the N_FBAP / N_TAP ratios? If yes, this should be mentioned as well.

**Answer: Thank you for this comment. We added the following sentence: "It should also be noted that we only counted particles with diameters between 1 and 12 µm, which means that results from studies with other particle diameter ranges might not be comparable." on line 310-311.**

l. 302 and 308-309: It is mentioned two times ‚that these differences also influence ratios between NFBAP and NTAP.' One occurrence is probably redundant.

**Answer: Thank you for noticing this. We removed the sentence on line 308.**

l. 513: ‚underestimates the bioaerosol concentration and correctly classifies biological material' -> ‚underestimates the bioaerosol concentration but correctly classifies biological material' ?

**Answer: Thank you we updated the "and" to a "but" on line 513.**

Fig. S5: There is probably a typo in the figure caption. It says: 'In addition to this, figure S6a seems to indicate that the increase in NFBAP scaled with the intensity of the rain event.'. I think it should be ‚…figure S5a seems to…'.

**Answer: Thank you, we updated it to say S5 instead of S6.**

Fig. S8 & S9, l. 507, l. 508: The figure numbers are probably S7 and S8, respectively.

**Answer: We had by mistake given the figures number S8 and S9 while it should correctly be S7 and S8. Thank you for pointing this out! We have updated the reference to these figures on lines 507 and 508.**

*Answers to referee #2 comments and remarks:*

Referee #2:
Line 70 of main manuscript -> Hund GmbH has in the meantime put a new model on the market. Please replace "BAA300 (BAA 500, Hund GmbH)" by "POMO-BAA500e (Helmut Hund GmbH, Germany)".

**Answer: Thank you for this. We have replaced the line accordingly see line 70.**

*Answers to editor remarks:*

Line 23: Please add "concentration" after "number".

**Answer: We have done this. See line 23.**

Line 90: Please delete (AT) as this is not consistently used along the text.

**Answer: We did this. See line 90.**

Line 120: Both terms were already defined in line 102.

**Answer: Thank you we removed this additional line on 120.**

Line 259: I suggest changing "This is due to that the" with "This is because the..." or with something equivalent.

**Answer: We updated the line to be "This is because the …" see line**

Line 282: Please delete "(Lieberherr et al., 2021)" as this is mentioned two lines above.

**Answer: We updated the sentence, see line 279.**

Line 287: N_TAP was already defined in L128.

**Answer: We removed the extra definition, see line 286.**

Lines 296-297: I think something is missing here. Please improve the grammar.

**Answer: Thank you for noticing this. We updated the sentence to the following "The seasonal average $N_{TAP}$ was between 3.42 and 4.96 cm$^{-3}$, which is higher than in most other similar studies where both $N_{FBAP}$ and $N_{TAP}$ have been measured." see line 296.**

Line 298: Please add a period after "UV-APS".

**Answer: We added the period, see line 298.**

Line 306 and along the text: Figures from the supplementary material must be called consecutively in the main text.

**Answer: Thank you for noticing this, we have renamed the figures and are referring to the accordingly now. See line 305.**

Line 311: "NTAP" and "NFBAP" need to be fixed.

**Answer: Thank you for noticing this. We have updated it accordingly. See line 311.**

Lines 451-452: I suggest replacing "To test the robustness of the analysis where a rain event was defined by the threshold of 0.5 mm h-1, we also applied other thresholds (including 1 mm h-1 and 2 mm h-1) but none of the thresholds used..." with "To test the robustness of the analysis a rain event was defined by the threshold of 0.5 mm h-1. Although we also applied other thresholds (including 1 mm h-1 and 2 mm h-1) none of the thresholds used..."

**Answer: Thank you, we updated the manuscript accordingly. See lines 450-453**

Line 518: Please add a period after "BioTrak".

**Answer: We added the period, see line 517.**

Lines 528-529: I think the authors forgot to delete this sentence.

**Answer: We did, thank you. We have now removed the line.**

Line 544: I am not sure if "prohibited" is completely correct. How about "inhibited"?

**Answer: We agree and updated the word to inhibited. See line 541.**

Figure 5: I suggest deleting (AT) in the caption and to replace "AT" with "air temperature" in the y-axis of panel b.

**Answer: Thank you, we updated it to be air temperature instead of AT both in the caption and on the y-axis of Figure 5b.**

Figure S2 is not mentioned in the main text.
There is no Figure S7. Therefore, Figure S8 and S9 should be S7 and S8, respectively. Fix this in both files.

**Answer: We updated S8 and S9 to be S7 and S8, respectively. We also added a line with reference to figure S2, see lines 228-229.**